# Block-Sample MAC-Bayes Generalization Bounds

**Matthias Frey and Jingge Zhu**
Department of Electrical and
Electronic Engineering
The University of Melbourne

**Michael C. Gastpar**
Laboratory for Information in Networked Systems
École polytechnique fédérale de Lausanne (EPFL)

## Abstract

We present a family of novel block-sample MAC-Bayes bounds (mean approximately correct). While PAC-Bayes bounds (probably approximately correct) typically give bounds for the generalization error that hold with high probability, MAC-Bayes bounds have a similar form but bound the expected generalization error instead. The family of bounds we propose can be understood as a generalization of an expectation version of known PAC-Bayes bounds. Compared to standard PAC-Bayes bounds, the new bounds contain divergence terms that only depend on subsets (or *blocks*) of the training data. The proposed MAC-Bayes bounds hold the promise of significantly improving upon the tightness of traditional PAC-Bayes and MAC-Bayes bounds. This is illustrated with a simple numerical example in which the original PAC-Bayes bound is vacuous regardless of the choice of prior, while the proposed family of bounds are finite for appropriate choices of the block size. We also explore the question whether high-probability versions of our MAC-Bayes bounds (i.e., PAC-Bayes bounds of a similar form) are possible. We answer this question in the negative with an example that shows that in general, it is not possible to establish a PAC-Bayes bound which (a) vanishes with a rate faster than $\mathcal{O}(1/\log n)$ whenever the proposed MAC-Bayes bound vanishes with rate $\mathcal{O}(n^{-1/2})$ and (b) exhibits a logarithmic dependence on the permitted error probability.

## 1 Introduction

Statistical learning theory studies the behavior of algorithms that infer a *hypothesis* $W$ from a *training set* $S$. This framework can be understood as a mathematical abstraction of real-world machine learning algorithms. For instance, if we want to analyze a neural network for image classification, $S$ would be a sequence of images with class labels and $W$ would be the set of all the weights and biases of the trained neural network. In this example, a standard way of finding $W$ given $S$ would be to run a stochastic gradient descent algorithm. Once such an algorithm is implemented, it defines a (possibly stochastic) mapping which (possibly randomly) assigns a hypothesis $W$ to any given training set $S$. We write this algorithm as a conditional probability distribution $P_{W|S}$. The analysis of the performance of such algorithms deals with the relationship of two fundamental quantities: The *empirical loss* describes how well the hypothesis given by the algorithm fits the training set $S$, while the *population loss* describes how well the hypothesis fits new data that may not be contained in $S$ but follows the same probability distribution that generated $S$. While the empirical loss of any given hypothesis can easily be calculated, the population loss is harder to deal with. It can, e.g., be estimated if an additional testing set is available that follows the same distribution as the training set, and while this is commonly done in practice, it carries many limitations. For instance, the testing set is usually obtained by partitioning the set of available labeled data (making the training set smaller) and if the algorithm is tweaked based on testing results, then technically, a new and independent testing set would be required to test the new algorithm. One fundamental question of statistical learning theory therefore is to understand and characterize the *generalization error*, which is defined as the difference between the empirical and population loss. If the generalization error is well understood, we can draw conclusions about the population loss from calculating the empirical loss.

More concretely, in the scenario we study in this paper, we assume that $S = (Z_1, \ldots, Z_n)$ is a sequence of length $n$ that is distributed i.i.d. according to a data generating distribution $P_Z$ on a set $\mathcal{Z}$, and the way in which we measure performance is described by a loss function $\ell : \mathcal{W} \times \mathcal{Z} \to [0, \infty)$. In the image classification example, each $Z_i$ would be an image and its associated classification label (the ground truth) and the loss function could be the classification loss, i.e., it maps a pair $(w, z)$ to 0 if the classification label produced by the neural network with weights $w$ and the image contained in $z$ matches the classification label contained in $z$ and to 1 otherwise. On the basis of this loss function, the empirical loss of $w$ under $s = (z_1, \ldots, z_n)$ is defined as $\hat{L}(w, s) := \frac{1}{n} \sum_{i=1}^{n} \ell(w, z_i)$, and the population loss of $w$ is defined as $L(w) := \mathbb{E}_{P_Z} \ell(w, Z)$.

PAC-Bayes bounds, initiated by McAllester (1999) and Shawe-Taylor & Williamson (1997), provide a framework of bounding the generalization error and thus give a learner the ability to draw conclusions about the population loss based on a computation of the empirical loss. They usually take the form

$$\forall \delta \in (0, 1] \; P_S \left( d(\mathbb{E}_{P_{W|S}} \hat{L}(W, S), \mathbb{E}_{P_{W|S}} L(W)) \geq \frac{D\left(P_{W|S} || Q_W\right) + I(n, d) + \log \frac{1}{\delta}}{n} \right) \leq \delta,$$
(1)

where:

- $P_S := P_Z^n$ is the product distribution that generates the training set $S$.

- $d$ is a *comparator function*. In this paper, we make the assumption (which is standard in the context of PAC-Bayes and related bounds) that $d$ is jointly convex in its arguments. For the left hand side of the bound to coincide with the generalization error, we would choose $d(r, s) := s - r$, but other notions of distance are possible here as well.

- $D\left(P_{W|S} || Q_W\right)$ is the Kullback-Leibler (KL) divergence between $P_{W|S}$ and some *prior* $Q_W$.

- $Q_W$, although called the *prior*, does not need to be a representation of prior belief in the traditional Bayesian sense, but is rather a parameter that can be arbitrarily chosen for the evaluation of the bound. It is a probability distribution on the hypothesis space $\mathcal{W}$, and the bound is valid for every choice of $Q_W$ that does not depend on $S$. Ideally, $Q_W$ would be chosen in such a way that it makes the right hand side as small as possible (and therefore making the bound as tight as possible) but in practice it is often chosen to be a probability distribution that is feasible to calculate with, such as a uniform or Gaussian distribution.

- $I(n, d)$ is a quantity that depends on the number of training samples $n$ and $d$ as well as the prior $Q$. Usually it contains a moment generating function. For instance, in (Germain et al., 2009, Theorem 2.1), this quantity is $I(n, d) := \log \mathbb{E}_{P_S Q_W} \exp\left(nd(\hat{L}(W, S), L(W))\right)$. Note that while $Q_W$ can be chosen for the sake of analysis as a probability distribution that is tractable to calculate with, $P_S$ may be unknown and potentially too complicated to calculate expectations over. Therefore, an upper bound for this expression (which is uniform for a reasonable large class of $P_S$) is sometimes used as $I(n, d)$ instead.

One interesting aspect of the PAC-Bayes bound is that the full information about the learning algorithm $P_{W|S}$ is explicitly used in the bound through the divergence term $D\left(P_{W|S} || Q_W\right)$, leading to a tight generalization error bound even for practical learning algorithms with large models such as deep neural networks, as illustrated in, e.g., Dziugaite & Roy (2017) and Pérez-Ortiz et al. (2021).

A variation of PAC-Bayes bounds that are not necessarily valid with high probability but instead bound the expected generalization error $\mathbb{E}_{P_{S,W}} d(\hat{L}(W, S), L(W))$ have been called MAC-Bayes bounds (mean approximately correct) by Grunwald et al. (2021). Such bounds can be derived from PAC-Bayes bounds as variations, but sometimes it is possible to give a MAC-Bayes bound that is tighter than the corresponding expectation version of the PAC-Bayes bound. An example for a MAC-Bayes version of (1) would be

$$\mathbb{E}_{P_S} d(\mathbb{E}_{P_{W|S}} \hat{L}(W, Z), \mathbb{E}_{P_{W|S}} L(W)) \leq \frac{\mathbb{E}_{P_S} D\left(P_{W|S} || Q_W\right) + I'(n, d)}{n},$$
(2)

where $I'(n, d)$ is possibly slightly distinct from $I(n, d)$ in (1) but also contains a moment-generating function and exhibits a similar behavior.

In this work, we study a generalization of the MAC-Bayes bound (2), which we call *block-sample MAC-Bayes bound*, taking the form

$$\mathbb{E}_{P_S} d(\mathbb{E}_{P_{W|S}} \hat{L}(W, Z), \mathbb{E}_{P_{W|S}} L(W)) \leq \frac{\sum_{j=1}^{J} \mathbb{E}_{P_{S_j}} D\left(P_{W|S_j} || Q_W\right) + I''(n, d, J)}{n}, \qquad (3)$$

where:

- The training data $S$ is partitioned into $J$ *blocks* $S_1, \ldots, S_J$.
- $P_{W|S_j}$, defined as
$$P_{W|S_j} := \mathbb{E}_{P_{S_1, \ldots, S_{j-1}, S_{j+1}, \ldots, S_J}} P_{W|S} \qquad (4)$$
is the distribution of $W$ conditional only on $S_j$. It is important to note here that $P_{W|S_j}$ does not denote an alternative algorithm that generates a hypothesis from block $S_j$ only, but rather is an averaged form of the algorithm $P_{W|S}$.
- $I''(n, d, J)$ is, similarly to $I(n, d)$ above, a moment-generating function. An important thing to note here is that in addition to being dependent on $n, d$, and $Q_W$, it also depends on the number of blocks $J$.

The focus of this paper is to introduce and investigate the properties of this new type of bounds, the optimal choice of block size, and what possibilities there exist to derive high-probability bounds of this type. Our finding shows that the obtained MAC-Bayes bounds can have right side terms that vanish at faster rates than that of the original PAC-Bayes bound or even vanish in some cases in which the original PAC-Bayes bound would be vacuous.

While ultimately the motivation for proposing a new type of bound is its potential to tighten existing PAC-Bayes based bounds for real-world learning algorithms such as the image classification example discussed above, substantial additional research will be necessary to solve many of the practical issues that arise when applying the bound to complex algorithms. In this paper, we limit ourselves to illustrating the promise of the bounds in the context of a very simple example, namely the estimation of a Gaussian mean under a truncated square loss. Specifically, in this example, $P_Z = \mathcal{N}(\mu, 1)$ (the normal distribution with mean $\mu$ and variance 1) and the loss function is given by

$$\ell(w, z) := K\left((w - z)^2\right), \quad K(x) := \begin{cases} x, & x \in [0, 1) \\ 1, & x \in [1, \infty). \end{cases}$$

The training algorithm $P_{W|S}$ is a Dirac distribution with probability mass at $\frac{1}{n} \sum_{i=1}^{n} Z_i$, i.e., the algorithm deterministically computes $W := \frac{1}{n} \sum_{i=1}^{n} Z_i$. It turns out that for deterministic learning algorithms like this one, the original PAC-Bayes bound (1) and its MAC-Bayes version (2) are vacuous (the right hand side is infinite) regardless of the choice of prior $Q_W$ while the block-sample bound (3) with choice $J > 1$ yields meaningful bounds.

We now briefly summarize the contribution of this paper.

- We prove a block-sample MAC-Bayes bound (Theorem 1) which generalizes the MAC-Bayes version of the original PAC-Bayes bound. In Corollary 1 we show how this general result can be used to obtain a bound on the generalization error by substituting a specific choice of distance function. We also illustrate what happens if binary KL divergence or difference function are substituted instead. While these bounds are suboptimal compared to Corollary 1, it is worth mentioning that substituting the difference function (Corollary 2) yields a bound with slightly wider applicability.
- The usefulness of the block-sample bounds is illustrated via an example in Section 4 where we show that the new bounds are in general tighter than the original PAC-Bayes bounds. The bound is also validated numerically. In Section 5, we generalize the scenario and illustrate how the growth behavior of the divergence term that appears in the bound influences the choice of $m$ that optimizes the bound order-wise.
- We explore the question of whether block-sample PAC-Bayes bounds are possible (i.e., a high probability version of the proposed block-sample MAC-Bayes bound). In Section 6, we answer this question in the negative, proving that in general, reasonably fast decaying bounds with a logarithmic dependence on the permissible error probability are not possible.

Like many other information-theoretic bounds (e. g. Bu et al. (2020) and Negrea et al. (2019)), the bounds we propose in this work depend on the distribution of the training data. This means that our bound can potentially be tighter than distribution-independent bounds in cases where at least partial statistical knowledge about the training data is available, but it comes with the drawback that the bound may not be computable in cases in which such knowledge is not available. In Section 7 we discuss the limitations that arise from this in more detail.

## 2 RELATED WORKS

Numerous PAC-Bayes bounds have been developed since its conception. For i.i.d. data, a framework was developed by Germain et al. (2009) and Bégin et al. (2016), where one class of PAC-Bayes bounds can be obtained systematically by choosing different comparator functions $d$. This framework allows us to recover well-known PAC-Bayes bounds from McAllester (1999) (improved by Maurer (2004)), Seeger (2002) and Catoni (2007), etc. More recent work on PAC-Bayes bounds also deals with non-i.i.d. data. For example, Alquier & Guedj (2018) established bounds for dependent, heavy-tailed observations, and Haddouche & Guedj (2022) considered a sequential model. PAC-Bayes bounds have recently received renewed interest partly due to their ability to provide tight generalization bounds for practical algorithms with large models. The idea of applying PAC-Bayes bounds to neural networks goes back to Langford & Caruana (2001). More recently, it has been shown by Dziugaite & Roy (2017) and Pérez-Ortiz et al. (2021) that when combined with suitable optimization techniques, PAC-Bayes bounds can generate very tight bounds. The recent survey by Alquier (2024) provides a comprehensive review of this topic.

MAC-Bayes bounds have appeared in the literature under various names, such as *integrated PAC-Bayes bounds*, *PAC-Bayes bounds in expectation*, or simply as *PAC-Bayes bounds*. Early appearances can, e.g., be found in Alquier (2006) and Catoni (2007), but they have also been of interest more recently in Grunwald et al. (2021). MAC-Bayes bounds have been studied in cases in which nontrivial PAC-Bayes bounds are not possible. A brief overview of MAC-Bayes bounds is included in the recent survey by Alquier (2024).

A related line of works appeared later in the information theory literature, starting with the works Russo & Zou (2016); Xu & Raginsky (2017). Results similar in spirit to PAC-Bayes bounds are presented under the name *information-theoretic bounds* on the generalization error. Works on information-theoretic bounds often focus on bounds in expectation, with many also proposing high-probability bounds. Examples of works that focus mostly on high-probability bounds are Esposito et al. (2021); Hellström & Durisi (2020). Further variations and refinements of information-theoretic bounds on generalization error include Asadi et al. (2018); Steinke & Zakynthinou (2020); Bu et al. (2020); Zhou et al. (2022); Wu et al. (2022; 2025). A recent survey by Hellström et al. (2025) provides an overview.

Our block-sample bound is inspired by the *individual-sample* information-theoretic bound due to Bu et al. (2020) in which the KL divergence terms are replaced by the mutual information of the individual samples contained in the training data and the hypothesis. The individual-sample bound is tighter than the original information-theoretic bound and has been applied to various learning scenarios, e.g., by Rodríguez-Gálvez et al. (2021); Harutyunyan et al. (2021). The work Harutyunyan et al. (2021) also considers bounds based on the mutual information of the hypothesis with $m$ training samples drawn uniformly at random. In the present work, in contrast, the training set is divided into blocks of size $m$ and the conditional distribution of the hypothesis given each of them shows up in the information quantities in our bounds. In the follow-up work Harutyunyan et al. (2022), the authors also give an impossibility result for high-probability statements which resembles Theorem 2 of the present paper, but is for the different system setup also used in Harutyunyan et al. (2021), is based on a fundamentally different idea, and intersects with our result only in the special case where the block size is $m = 1$. For $m > 1$, the impossibility result in Harutyunyan et al. (2022) is not applicable. Wu et al. (2024) also study PAC-Bayes bounds based on a separation of the training sample into blocks, however, their bound differs from the ones proposed in this paper in that it has a recursive structure based on sequences of prior and posterior distributions, which makes it not directly comparable to our bound. PAC-Bayes bounds involving individual samples have been considered in some recent work including Zhou et al. (2023); Hellström & Durisi (2022). The recent works Foong et al. (2021); Hellström & Guedj (2024) studied the tightness of original PAC-Bayes bounds

with different choices of the comparator function $d$, which are relevant to our current study. To the best knowledge of the authors, this paper is the first work that presents block-based MAC-Bayes bounds under a general framework, considers the optimization of the bound, and explores possible PAC-Bayes versions of the bound.

## 3 BLOCK-SAMPLE MAC-BAYES BOUNDS

In this section, we propose the general version of the block-sample MAC-Bayes bound along with specializations to some commonly used comparator functions $d$. In this bound, the training set $S$ is partitioned into blocks $S_j := Z_{(j-1)m+1:jm} := (Z_{(j-1)m+1}, \ldots, Z_{jm})$.

**Theorem 1** (Block-sample MAC-Bayes bounds). *Let $m \in [n] := \{1, \ldots, n\}$, assume that $n$ is an integer multiple of $m$, and let $S' = (Z'_1, \ldots, Z'_m)$ where $Z'_1 \ldots, Z'_m$ are i.i.d. drawn from $P_Z$. Assume that for $\lambda' \in (0, b)$ and some distribution $Q_W$ over $\mathcal{W}$, it holds that*

$$\mathbb{E}_{P_{S'} Q_W} \exp \left( \lambda' d \left( \frac{1}{m} \sum_{i=1}^m \ell(W, Z'_i), L(W) \right) \right) \leq \Phi_m(\lambda') \tag{5}$$

*for some function $\Phi_m : (0, b) \to (0, \infty)$. Then for any $\lambda \in (0, bn/m)$, it holds that*

$$\mathbb{E}_{P_S} d \left( \mathbb{E}_{P_{W|S}} \hat{L}(W, S), \mathbb{E}_{P_{W|S}} L(W) \right) \leq \frac{\frac{n}{m} \log \Phi_m \left( \frac{\lambda m}{n} \right) + \sum_{j=1}^J \mathbb{E}_{P_{S_j}} D \left( P_{W|S_j} || Q_W \right)}{\lambda}, \tag{6}$$

*where $J := n/m$.*

*Proof.* The proof uses the Donsker-Varadhan variational representation of KL divergence to achieve a change of measure from the marginalized version of the algorithm $P_{W|S_j}$ to the prior $Q_W$. Specifically, we argue

$$\mathbb{E}_{P_S} d \left( \mathbb{E}_{P_{W|S}} \hat{L}(W, S), \mathbb{E}_{P_{W|S}} L(W) \right)$$

$$\overset{(a)}{\leq} \mathbb{E}_{P_S} \mathbb{E}_{P_{W|S}} d \left( \hat{L}(W, S), L(W) \right)$$

$$= \mathbb{E}_{P_S} \mathbb{E}_{P_{W|S}} d \left( \frac{1}{J} \sum_{j=1}^J \frac{1}{m} \sum_{i=(j-1)m+1}^{jm} \ell(W, Z_i), L(W) \right)$$

$$\overset{(a)}{\leq} \mathbb{E}_{P_S} \mathbb{E}_{P_{W|S}} \left( \frac{1}{J} \sum_{j=1}^J d \left( \frac{1}{m} \sum_{i=(j-1)m+1}^{jm} \ell(W, Z_i), L(W) \right) \right)$$

$$\overset{(b)}{=} \sum_{j=1}^J \mathbb{E}_{P_{S_j}} \left( \mathbb{E}_{P_{S_1, \ldots, S_{j-1}, S_{j+1}, \ldots, S_J}} \mathbb{E}_{P_{W|S}} \left( \frac{1}{J} d \left( \frac{1}{m} \sum_{i=(j-1)m+1}^{jm} \ell(W, Z_i), L(W) \right) \right) \right)$$

$$\overset{(4)}{=} \sum_{j=1}^J \mathbb{E}_{P_{S_j}} \left( \mathbb{E}_{P_{W|S_j}} \left( \frac{1}{J} d \left( \frac{1}{m} \sum_{i=(j-1)m+1}^{jm} \ell(W, Z_i), L(W) \right) \right) \right)$$

$$= \sum_{j=1}^J \mathbb{E}_{P_{S_j}} \left( \frac{1}{\lambda} \mathbb{E}_{P_{W|S_j}} \left( \frac{\lambda}{J} d \left( \frac{1}{m} \sum_{i=(j-1)m+1}^{jm} \ell(W, Z_i), L(W) \right) \right) \right)$$

$$\overset{(c)}{\leq} \sum_{j=1}^J \mathbb{E}_{P_{S_j}} \left( \frac{1}{\lambda} \left( \log \mathbb{E}_{Q_W} \exp \left( \frac{\lambda}{J} d \left( \frac{1}{m} \sum_{i=(j-1)m+1}^{jm} \ell(W, Z_i), L(W) \right) \right) \right. \right.$$

$$\left. \left. + D \left( P_{W|S_j} || Q_W \right) \right) \right)$$

$$\overset{(d)}{\leq} \sum_{j=1}^{J} \frac{\log \mathbb{E}_{P_{S_j}} \mathbb{E}_{Q_W} \exp\left(\frac{\lambda}{J} d\left(\frac{1}{m}\sum_{i=(j-1)m+1}^{jm} \ell(W,Z_i), L(W)\right)\right) + \mathbb{E}_{P_{S_j}} D\left(P_{W|S_j}||Q_W\right)}{\lambda}$$

$$\overset{(5)}{\leq} \sum_{j=1}^{J} \frac{\log \Phi_m\left(\frac{\lambda}{J}\right) + \mathbb{E}_{P_{S_j}} D\left(P_{W|S_j}||Q_W\right)}{\lambda}$$

$$= \frac{J\log \Phi_m\left(\frac{\lambda}{J}\right) + \sum_{j=1}^{J} \mathbb{E}_{P_{S_j}} D\left(P_{W|S_j}||Q_W\right)}{\lambda},$$

where both steps labeled (a) are applications of Jensen's inequality which make use of the convexity of $d$, (b) is due to the Fubini-Tonelli theorem, (c) is an application of the Donsker-Varadhan inequality

$$\mathbb{E}_{P_{W|S_j}} g(W) \leq \log \mathbb{E}_{Q_W} \exp(g(W)) + D\left(P_{W|S_j}||Q_W\right)$$

with

$$g(W) := \frac{\lambda}{J} d\left(\frac{1}{m}\sum_{i=(j-1)m+1}^{jm} \ell(W,Z_i), L(W)\right),$$

and (d) uses Jensen's inequality with the concavity of the logarithm. $\square$

**Remark 1.** *Condition (5) can be understood as assuming an upper bound on the moment generating function of the loss function. As can be seen in the proofs of Corollaries 1 and 2, it is always satisfied for bounded and subgaussian losses.*

**Remark 2** (Data dependent priors). *It can be verified in the proof of Theorem 1 that the result also holds if $Q_W$ is replaced in (5) with a data dependent prior $Q_{W|S'}$. The prior will then appear in (6) as $Q_{W|S_j}$. In the remainder of this paper, we will focus on the case that $Q_W$ does not have any dependence on data.*

Theorem 1 can be specialized by substituting various choices for $d$. In this paper, we look at bounds in terms of the comparators $d(r,s) := C_\beta(r,s) := -\log\left(1 - \left(1 - \exp(-\beta)\right)s\right) - \beta r$ (this is the *Catoni function* which was introduced as a comparator by Catoni (2007)), $d(r,s) := \mathrm{kl}\left(r||s\right) := s\log\frac{s}{r} + (1-s)\log\frac{1-s}{1-r}$ where we define $0\log 0 := 0$ by convention (this is the *binary KL function* which is defined for $r,s \in [0,1]$), and $d(r,s) := s - r$ (the difference function). If the result is in terms of the difference function, it directly bounds the expected generalization error $\mathrm{gen} := \mathbb{E}_{P_{S,W}}\left(L(W) - \hat{L}(W,Z)\right)$.

The following corollary shows how Theorem 1 can be used to upper bound the generalization error of a learning algorithm in the case of bounded loss. It follows from Theorem 1 by substituting $d(r,s) := C_\beta(r,s)$. Full details of the proof can be found in Appendix A.

**Corollary 1.** *Let $\ell(w,z) \in [0,1]$ for all $w \in \mathcal{W}, z \in \mathcal{Z}$. Then for any $\beta > 0$, we have*

$$\mathbb{E}_{P_S} C_\beta\left(\mathbb{E}_{P_{W|S}}\hat{L}(W,S), \mathbb{E}_{P_{W|S}}L(W)\right) \leq \frac{1}{n}\sum_{j=1}^{J} \mathbb{E}_{P_{S_j}} D\left(P_{W|S_j}||Q_W\right). \tag{7}$$

*Furthermore,*

$$\mathrm{kl}\left(\mathbb{E}_{P_{S,W}}\hat{L}(W,S)||\mathbb{E}_{P_{S,W}}L(W)\right) \leq \frac{1}{n}\sum_{j=1}^{J} \mathbb{E}_{P_{S_j}} D\left(P_{W|S_j}||Q_W\right) \tag{8}$$

$$\mathrm{gen} \leq \sqrt{\frac{1}{4n}\sum_{j=1}^{J} \mathbb{E}_{P_{S_j}} D\left(P_{W|S_j}||Q_W\right)}. \tag{9}$$

**Remark 3.** *Under the assumption $\ell(w,z) \in [0,1]$ for all $w \in \mathcal{W}, z \in \mathcal{Z}$, we can of course also directly substitute $d(r,s) := \mathrm{kl}\left(r||s\right))$ in Theorem 1 in an attempt to obtain a bound akin to (8). This would give us (for full details see Appendix A)*

$$\mathbb{E}_{P_S}\mathrm{kl}\left(\mathbb{E}_{P_{W|S}}\hat{L}(W,S)||\mathbb{E}_{P_{W|S}}L(W)\right) \leq \frac{\log(2\sqrt{m})}{m} + \frac{1}{n}\sum_{j=1}^{J} \mathbb{E}_{P_{S_j}} D\left(P_{W|S_j}||Q_W\right). \tag{10}$$

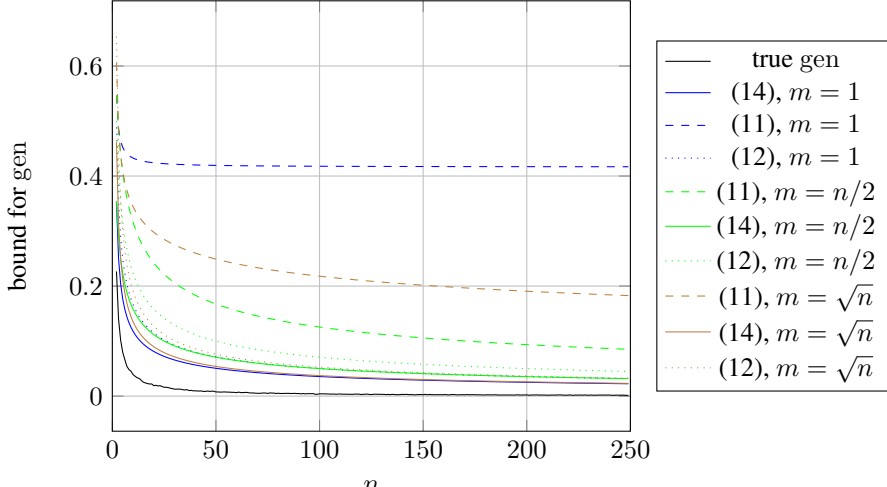

Figure 1: Comparison of true generalization error and theoretical bounds for the example in Section 4 with $\mu = 1/2$. The solid blue curve shows the optimal bound (14) with optimal choice for $m$.

*However, it is easy to see that after an application of Jensen's inequality this yields a bound that is less tight than* (8) *due to the presence of the additional summand* $\log(2\sqrt{m})/m$. *In terms of generalization error, this would yield the suboptimal bound*

$$\text{gen} \leq \frac{1}{2}\sqrt{\frac{\log(2\sqrt{m})}{m} + \frac{1}{n}\sum_{j=1}^{J} \mathbb{E}_{P_{S_j}} D\left(P_{W|S_j} || Q_W\right)}. \tag{11}$$

We conclude this section with a corollary that shows how Theorem 1 can be applied in case the loss is not necessarily bounded. For this, we replace the boundedness assumption with the less general assumption that the loss is subgaussian and substitute $d(r, s) = s - r$ in Theorem 1 to obtain the following result. The further details of the proof can be found in Appendix A.

**Corollary 2.** *Assume that the loss function $\ell(w, Z)$ is $\sigma^2$-subgaussian for any $w$ under the distribution $P_Z$, namely,*

$$\mathbb{E}_{P_Z} \exp(\lambda'(\ell(w, Z) - L(w))) \leq \exp(\sigma^2 \lambda'^2/2)$$

*for all $\lambda' \in \mathbb{R}$. Then for any $\lambda > 0$ and a prior distribution $Q_W$ over $\mathcal{W}$, it holds that*

$$\text{gen} \leq \sqrt{\frac{2\sigma^2}{n}\sum_{j=1}^{J} \mathbb{E}_{P_{S_j}} D\left(P_{W|S_j} || Q_W\right)}. \tag{12}$$

*The same result holds if we assume the (weaker) condition $\mathbb{E}_{P_Z Q_W} \exp\left(\lambda'(\ell(W, Z) - L(W))\right) \leq \exp(\sigma^2 \lambda'^2/2)$, namely that $\ell(W, Z) - L(W)$ is $\sigma^2$-subgaussian under the product distribution $P_Z Q_W$.*

## 4 EXAMPLE: GAUSSIAN MEAN ESTIMATION WITH TRUNCATED LOSS FUNCTION

In this section, we explore the simple example toy example for a learning scenario which was introduced in Section 1. The example shows that the proposed block-sample MAC-Bayes bounds can yield meaningful convergence guarantees in cases in which the original PAC-Bayes bound would be vacuous. In the following, we will use the Landau symbols $\mathcal{O}, \Theta$, and $o$ according to their standard definitions.

Recall that $Z_i \sim \mathcal{N}(\mu, 1)$ for some (unknown) $\mu \in (0, 1)$, $W := \frac{1}{n}\sum_{i=1}^{n} Z_i$, and

$$\ell(w, z) := K\left((w - z)^2\right), \quad K(x) := \begin{cases} x, & x \in [0, 1) \\ 1, & x \in [1, \infty). \end{cases}$$

| $\mathbb{E}_{P_S}\hat{L}(W,S)$ | gen with $m=\Theta\left(n^\alpha\right)$ | $\alpha^*$ | gen with $m=\Theta\left(n^{\alpha^*}\right)$ |
|---|---|---|---|
| any | $\mathcal{O}\left(n^{\frac{\alpha(\gamma-1)-1}{2}}\right)$ | $\mathbb{1}_{\gamma<1}$ | $\mathcal{O}\left(n^{\frac{\min(\gamma-1,0)-1}{2}}\right)$ |
| $\mathcal{O}\left(n^{-\varepsilon}\right)$ | $\mathcal{O}\left(n^{\alpha(\gamma-1)-1}\right)+\mathcal{O}\left(n^{-\varepsilon}\log n\right)$ | $\mathbb{1}_{\gamma<1}$ | $\mathcal{O}\left(n^{\min(\gamma-1,0)-1}\right)+\mathcal{O}\left(n^{-\varepsilon}\log n\right)$ |

Table 1: Convergence rate of gen depending on the growth behavior of $m$ under assumption (15).

This loss function has, e.g., been analyzed and applied to practical ML algorithms by Le & Zach (2021).

To understand the choice of $m$ for the KL-divergence term, we choose the set $S_j$ to be the $j$-th batch with $m$ samples, namely $S_j = Z_{(j-1)m+1:jm}$ and define $\bar{S}_j$ to be the sum of these elements $\bar{S}_j := \sum_{i=(j-1)m+1}^{jm} Z_i$. It can be seen that we have $P_{W|S_j} = \mathcal{N}\left(\mu(n-m)/n + \bar{S}_j/n, (n-m)/n^2\right)$. Choosing the prior $Q_W := \mathcal{N}\left(\mu, (n-m)/n^2\right)$ gives us

$$\mathbb{E}_{P_S} D\left(P_{W|S_j}||Q_W\right) \overset{(a)}{=} \mathbb{E}_{P_S}\left(\frac{n^2}{2(n-m)}\left(\frac{\bar{S}_j}{n}-\frac{\mu m}{n}\right)^2\right) = \frac{\mathbb{E}_{P_S}\left(\left(\bar{S}_j-\mu m\right)^2\right)}{2(n-m)} \overset{(b)}{=} \frac{m}{2(n-m)},$$
(13)

where in (a) we have used the KL divergence formula from (Gil, 2011, Table 2.3) and (b) follows because $\bar{S}_j$ follows the distribution $\mathcal{N}(\mu m, m)$. Substituting (13) in (9), we obtain

$$\text{gen} \leq \sqrt{\frac{1}{4n}\cdot\frac{n}{m}\cdot\frac{m}{2(n-m)}} = \frac{1}{2}\sqrt{\frac{1}{2(n-m)}}.$$
(14)

The original PAC-Bayes bound, which would correspond to the choice $m=n$, can clearly be seen to be vacuous in this case. However, any other choice of block size yields a bound which decays as $\mathcal{O}(n^{-\frac{1}{2}})$, with $m=1$ being the optimal choice. We show the bound (14) in Figure 1 for various choices of block size $m$ and plot for illustration also the suboptimal bounds for gen that can be derived from (11) and (12) in a similar way. It can be seen in the plots that the bound (14) is not overly sensitive to suboptimal choices of $m$ (as long as $m \neq n$) and that if adjusting $m$ appropriately, the suboptimal bounds derived from (11) and (12) will also decay reasonably as $n$ grows.

**Remark 4.** *While the choice of $Q_W$ we have used for the evaluation of this example is somewhat arbitrary, it is worth noting that the bound for the case $m=n$ (corresponding to the original PAC-Bayes bound) is vacuous for every possible choice of $Q_W$. For a detailed argument, see Appendix B.*

## 5 OPTIMIZING BLOCK-SAMPLE MAC-BAYES BOUNDS

The example from the previous section illustrates that the right choice of $m$ depends on the behavior of the divergence term $\mathbb{E}_{P_S}\sum_{j=1}^J D\left(P_{W|S_j}||Q_W\right)$, which needs to be evaluated on a case-by-case basis. In this section, we illustrate this finding order-wise in more generality under the assumption that the summands of the divergence term satisfy

$$\mathbb{E}_{P_S} D\left(P_{W|S_j}||Q_W\right) \leq \frac{\mathcal{O}(m^\gamma)}{\Theta(n)}$$
(15)

for some $\gamma \geq 0$ and all $S_j$. Of course, the assumption in (15) needs to be verified for the problem at hand. For instance it can be seen in (13) that for the example in Section 4, (15) holds with $\gamma=1$ as long as $m \neq n$.

In Table 1, we show bounds for the generalization gap $\text{gen} = \mathbb{E}_{P_{S,W}}(L(W)-\hat{L}(W,Z))$ that can be obtained from Corollary 1 under assumption (15). The notation $\mathbb{1}_\varphi$ which appears in the table is the indicator function, defined to take the value 1 when the condition $\varphi$ is satisfied and the value 0 otherwise. If nothing is known about the behavior of the empirical loss $\mathbb{E}_{P_S}\hat{L}(W,S)$, only the bound given in the first row applies. If, on the other hand, it is known that the empirical loss vanishes as

$\mathbb{E}_{P_S} \hat{L}(W, S) = \mathcal{O}\left(n^{-\varepsilon}\right)$ where $\varepsilon > 0$, there is an additional trick which enables us to use the bound in the second row. This bound can, depending on the value of $\varepsilon$, be significantly tighter than the bound that does not use this assumption. For details on how these bounds are calculated, we refer the reader to Appendix C.

The bounds are given for choices of block size assumed to behave as $m = \Theta(n^\alpha)$, where $\alpha \in [0, 1]$ is a parameter that can be optimized. Correspondingly, we indicate the optimal value $\alpha^*$ of $\alpha$ along with the resulting order-wise behavior of the generalization bound. It can be seen in the table that the example from Section 4 sits exactly at the transition point of a dichotomy: For $\gamma < 1$, any constant choice for $m$ is optimal while for $\gamma > 1$, the block size $m$ should grow linearly with $n$. This would include the choice $m = n$ as well as choices such as $m = n/2$. It is worth noting here that as can be seen in the example of Section 4, (15) is in some cases only satisfied if $m \neq n$. In these cases, the original PAC-Bayes bound $(m = n)$ is not applicable (for instance, in our example it would be vacuous) but the block-sample approach still allows us to achieve the order-wise optimal tradeoff point with a choice such as $m = n/2$.

## 6 ON THE (IM)POSSIBILITY OF BLOCK-SAMPLE PAC-BAYES BOUNDS

In this section, we explore the possibility of transforming the block-sample MAC-Bayes bound into a version that holds with high probability, and thus obtaining a PAC-Bayes bound. Specifically, we are interested in the following question: *Is it possible to find a meaningful PAC-Bayes bound for every learning scenario for which Theorem 1 yields a MAC-Bayes bound that converges to $0$ (reasonably fast)?* For the interpretation of *meaningful PAC-Bayes bound*, we will focus on the question whether bounds of the following form exist:

$$P_S \left( \mathbb{E}_{P_{W|S}} d\left(\hat{L}(W, S), L(W)\right) \leq A_n + B_n \cdot f\left(\frac{1}{\delta}\right) \right) \geq 1 - \delta, \qquad (16)$$

where $A_n$ converges to $0$ as $n \to \infty$, $f$ is a function that grows slowly, and $B_n$ converges to $0$ reasonably fast as $n \to \infty$. Note that the right hand side of the original PAC-Bayes bound (1) is of the same form as the right hand side in (16) with $f = \log$ and $B_n = 1/n$. We show next that there exists a learning scenario with the following properties: (a) Theorem 1 gives a MAC-Bayes bound with a square root convergence, but (b) any PAC-Bayes bound of the form (16) which is valid for this learning scenario needs to either have a fast growing $f$ or a slowly converging $B_n$ (see the theorem statement for the technical meaning of fast growing and slowly converging). For this, we focus on the comparator function $d(r, s) = s - r$ in which case

$$\mathbb{E}_{P_S} d\left(\mathbb{E}_{P_{W|S}} \hat{L}(W, S), \mathbb{E}_{P_{W|S}} L(W)\right) = \text{gen} = \mathbb{E}_{P_{S,W}}\left(L(W) - \hat{L}(W, S)\right)$$

corresponds to the most commonly used definition of the generalization gap.

**Theorem 2.** *Let $m$ be chosen in dependence of $n$ such that $m/n \to 0$ as $n \to \infty$. Then there exist a loss function $\ell$ with values bounded in $[0, 1]$, a sample distribution $P_Z$, and a learning algorithm $P_{W|S}$ with the following properties:*

1. *The assumptions of Theorem 1 are satisfied with the choice $d(r, s) = s - r$.*

2. *There exists $Q_W$ such that the right hand side of (6) vanishes in order $\mathcal{O}(n^{-1/2})$ as $n \to \infty$.*

3. *For every nondecreasing function $f : [0, \infty) \to \mathbb{R}$, for every sequence $A_n$ that vanishes as $n \to \infty$, for every sequence $B_n$ in $o(1/f(n \log n))$, and for every sufficiently large $n$, there is $\delta \in [0, 1]$ such that*

$$P_S \left( \mathbb{E}_{P_{W|S}} \left(L(W) - \hat{L}(W, S)\right) > A_n + B_n \cdot f\left(\frac{1}{\delta}\right) \right) > \delta. \qquad (17)$$

The full proof of Theorem 2 is relegated to Appendix D. In it, we explicitly construct a learning scenario with the following property: With a small probability over $P_S$, the learning algorithm dramatically overfits to the training sample. With the remaining (larger) probability, it outputs a

hypothesis that has zero loss (both empirical and in population). We then show that items 1, 2, and 3 hold for this learning scenario, giving an explicit choice of $Q_W$ for item 2.

Items 1 and 2 in conjunction with Theorem 1 tell us that there exists a learning scenario in which the bound (6) holds with a right hand side of the order $\mathcal{O}(n^{-1/2})$. Item 3 tells us that in the same learning scenario, a bound of the form (16) cannot hold with a slowly growing $f$ and a fast converging $B_n$ at the same time (observe that (17) is the logical opposite of (16) with the difference function substituted for $d$).

We conclude this section with two remarks that approach the question of whether block-sample PAC-Bayes bounds are possible from slightly different angles.

**Remark 5.** *If $d$ takes only nonnegative values[1], then (6) and the Markov bound imply the following high-probability bound: For every $\delta \in (0,1)$, with probability at least $1 - \delta$ over $P_S$,*

$$d\left(\mathbb{E}_{P_{W|S}}\hat{L}(W,S), \mathbb{E}_{P_{W|S}}L(W)\right) < \frac{\frac{n}{m}\log\Phi_m\left(\frac{\lambda m}{n}\right) + \sum_{j=1}^{J}\mathbb{E}_{P_{S_j}}D\left(P_{W|S_j}||Q_W\right)}{\lambda\delta}.$$

*This bound has the form (16) with $A_n = 0$ and, depending on the learning scenario at hand, quite possibly with a term $B_n$ that converges to 0 reasonably fast. However, $f$ is the identity function here, which is much worse than the original PAC-Bayes bound (1) where we have $f = \log$.*

**Remark 6.** *While Theorem 2 conclusively shows that an exponential high-probability version of the MAC-Bayes bound (6) is not possible in general, it strictly speaking leaves open the possibility that a block-sample version of the original PAC-Bayes bound (1) of the following form might exist:*

$$\mathbb{P}\left(d(\hat{L}, L) \geq \frac{\sum_{j=1}^{J}D\left(P_{W|S_j}||Q_W\right) + I(n,d) + \log\frac{1}{\delta}}{n}\right) \leq \delta \qquad (18)$$

*The main difference compared to (6) is that the expectation operator in front of the divergence terms $D\left(P_{W|S_j}||Q_W\right)$ is absent in (18). However, if one inspects the proof of Theorem 2, it is clear that in the error event evaluated in (17) (for a detailed derivation see Appendix E), we have*

$$\Delta_{\text{inst}} := \sum_{j=1}^{J}D\left(P_{W|S_j}||Q_W\right) = \mathcal{O}(\log n), \qquad (19)$$

*so the term $\Delta_{\text{inst}}/n$ that appears in (18) vanishes at rate $\mathcal{O}(n^{-1}\log n)$. Therefore it is clear from item 3 of Theorem 2 that no bound of the form (18) with $I(n,d)/n \to 0$ as $n \to \infty$ can hold.*

## 7 CONCLUSIONS AND LIMITATIONS

We have presented a family of block-sample MAC-Bayes bounds which are generally tighter than MAC-Bayes versions of classical PAC-Bayes bounds, and we have shown that they cannot in general be established as PAC (i.e., high-probability) versions with logarithmic dependence on the error probability. One important further limitation of these bounds is their dependence on the data distribution $P_Z$ via the divergence term $D\left(P_{W|S_j}||Q_W\right)$. In this sense, they cannot be applied immediately if we do not have any information about the data distribution $P_Z$. Two comments are in order regarding this limitation: **(a)** Given certain basic information about the data distribution, block-sample MAC-Bayes bounds can give useful results without requiring full knowledge of the data distribution $P_Z$. In fact, as we show in Section 5, as long as we have enough knowledge of the data distribution to be able to establish an upper bound of the form (15), the block-sample MAC-Bayes bound can be applied. **(b)** More broadly, almost all information-theoretic bounds on generalization error depend on the data distribution as the mutual information term $I(W;S)$ (or variations thereof) depends on the data distribution (see, e.g., Hellström et al. (2025) for a recent survey). In such cases, to make the bounds useful, the mutual information term is further upper bounded by using knowledge of the data and learning algorithm. This is done by, e.g., Bu et al. (2020) and Negrea et al. (2019). A similar approach can be taken for the block-sample PAC-Bayes bounds by further bounding the divergence term $D\left(P_{W|S_j}||Q_W\right)$ with properties of the learning algorithm. We consider this an interesting future research direction.

---

[1]Of the three comparator functions studied in this paper, this applies only to the KL divergence. However, other comparator functions that satisfy this assumption are readily available, for instance the absolute difference function $d(r,s) = |r - s|$ or the square distance function $d(r,s) = (r - s)^2$.

ACKNOWLEDGMENTS

The work in this manuscript was partially supported by the Swiss National Science Foundation under Grant 200364 and by the Australian Research Council under Project DP23010149..

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

# A  PROOF DETAILS FOR COROLLARIES AND REMARKS IN SECTION 3

## A.1  PROOF OF COROLLARY 1

The proof uses similar techniques as in Foong et al. (2021) and Maurer (2004), and is given here for completeness. We will make use of the following lemma.

**Lemma 1** (Maurer (2004), Lemma 3). *Let $X_1, \ldots, X_k$ be i.i.d. random variables taking values in $[0, 1]$ and $X_1', \ldots, X_k'$ i.i.d. Bernoulli random variables such that $\mathbb{E}X_1 = \mathbb{E}X_1'$. For any convex function $f : [0, 1]^k \to \mathbb{R}$, it holds that*
$$\mathbb{E}f(X_1, \ldots, X_k) \leq \mathbb{E}f(X_1', \ldots, X_k').$$

For any fixed $w$, denote $X_i := \ell(w, Z_i')$ for all $i = 1, \ldots, m$ and use $X$ to denote the tuple $(X_1, \ldots, X_m)$. Define $X'$ to be a tuple of $m$ i.i.d. Bernoulli random variables such that $\mathbb{E}X_i = \mathbb{E}X_i'$ for all $i$. Let $f : X \mapsto \exp(\lambda' C_\beta(\sum_{i=1}^m X_i, L(w)))$. Thus, we obtain

$$\mathbb{E}_{P_{S'}} \exp\left(\lambda' C_\beta\left(\frac{1}{m}\sum_{i=1}^m \ell(w, Z_i'), L(w)\right)\right)$$

$$\overset{(a)}{\leq} \mathbb{E}_{P_{X'}} \exp\left(\lambda' C_\beta\left(\frac{1}{m}\sum_{i=1}^m X_i', L(w)\right)\right)$$

$$= \frac{1}{\left(1 + L(w)\left(\exp(-\beta) - 1\right)\right)^{\lambda'}} \mathbb{E}_{P_{X'}} \exp\left(-\frac{\beta\lambda'}{m}\sum_{i=1}^m X_i'\right)$$

$$\overset{(b)}{=} \frac{\left(1 + L(w)\left(\exp(-\beta\lambda'/m) - 1\right)\right)^m}{\left(1 + L(w)\left(\exp(-\beta) - 1\right)\right)^{\lambda'}},$$

where (a) is by Lemma 1 and in (b) we have used the well-known expression for the moment generating function of the binomial distribution. This upper bound is clearly equal to 1 if $\lambda' = m$, so (5) is satisfied for $\lambda' = m$ with $\Phi_m(m) = 1$. Consequently, an application of Theorem 1 with $\lambda = n$ yields (7).

To obtain (8) from (7), we observe that by (Foong et al., 2021, Lemma E.1),

$$\text{kl}\left(\mathbb{E}_{P_{S,W}}\hat{L}(W, S) \| \mathbb{E}_{P_{S,W}} L(W)\right) = \sup_{\beta \in (0,\infty)} C_\beta\left(\mathbb{E}_{P_{S,W}}\hat{L}(W, S), \mathbb{E}_{P_{S,W}} L(W)\right).$$

Finally, we can obtain (9) from (8) using Pinsker's inequality

$$\text{gen} \leq \frac{1}{2}\sqrt{\text{kl}\left(\mathbb{E}_{P_{S,W}}\hat{L}(W, S) \| \mathbb{E}_{P_{S,W}} L(W)\right)}.$$

## A.2  CALCULATION DETAILS FOR REMARK 3

We again fix $w \in \mathcal{W}$ and use Lemma 1 with $X_i := \ell(w, Z_i')$, corresponding Bernoulli variables $X_i'$, and $f : X \mapsto \exp\left(\lambda'\text{kl}\left(\sum_i X_i/m \| L(w)\right)\right)$. It can be checked easily that $f$ is convex. For any $w$, we have

$$\mathbb{E}_{P_{S'}}\left(\exp\left(\lambda'\text{kl}\left(\frac{1}{m}\sum_{i=1}^m \ell(w, Z_i') \| L(w)\right)\right)\right)$$

$$\overset{(a)}{\leq} \mathbb{E}_{P_{S'}}\left(\exp\left(\lambda'\text{kl}\left(\frac{1}{m}\sum_{i=1}^m X_i' \| L(w)\right)\right)\right)$$

$$= \sum_{k=0}^m P_{X'}\left(\sum_{i=1}^m X_i' = k\right)\exp\left(\lambda'\text{kl}\left(\frac{k}{m} \| L(w)\right)\right)$$

$$\leq \sup_{r \in [0,1]} \sum_{k=0}^m \text{Bin}(k; m, r)\exp\left(\lambda'\text{kl}\left(\frac{k}{m} \| r\right)\right)$$

where (a) follows from Lemma 1 and $\text{Bin}(k; m, r)$ denotes the pmf of a binomial random variable with $m$ trials and success probability $r$, evaluated at $k$. Using the result by (Maurer, 2004, Theorem 1), the last term is upper bounded by $2\sqrt{m}$ when choosing $\lambda' = m$. (Maurer, 2004, Theorem 1) states the assumption $m \geq 8$, however, as noted at the end of the proof of (Germain et al., 2015, Lemma 19), this assumption is not necessary as the remaining cases can be verified computationally. This means that (5) holds with $\Phi_m(m) := 2\sqrt{m}$. So an application of Theorem 1 with $\lambda := n$ yields

$$\mathbb{E}_{P_S} \text{kl}\left(\mathbb{E}_{P_{W|S}} \hat{L}(W, S) \| \mathbb{E}_{P_{W|S}} L(W)\right) \leq \frac{\frac{n}{m} \log(2\sqrt{m}) + \sum_{j=1}^{J} \mathbb{E}_{P_{S_j}} D\left(P_{W|S_j} \| Q_W\right)}{n}$$

from which (10) follows.

### A.3 PROOF OF COROLLARY 2

The assumption on the moment-generating function in the corollary straightforwardly implies

$$\mathbb{E}_{P_{S'}} \exp\left(\lambda'\left(\frac{1}{m} \sum_{i=1}^{m} \ell(w, Z_i') - L(w)\right)\right) \leq \exp\left(\frac{\sigma^2 \lambda'^2}{2m}\right)$$

for all $\lambda' \in \mathbb{R}$ and any $w$. Choosing $d(s, r) := r - s$, this then implies that for any $\lambda > 0$

$$\mathbb{E}_{P_{Z'} Q_W} \exp\left(\lambda d\left(\frac{1}{m} \sum_{i=1}^{m} \ell(W, Z_i'), L(W)\right)\right)$$

$$= \mathbb{E}_{P_{Z'} Q_W} \exp\left(-\lambda\left(\frac{1}{m} \sum_{i=1}^{m} \ell(W, Z_i') - L(W)\right)\right)$$

$$\leq \exp\left(\frac{\sigma^2 \lambda^2}{2m}\right)$$

$$=: \Phi_m(\lambda).$$

Furthermore, the above inequality also holds when the weaker condition in the corollary is assumed by noticing that $\mathbb{E}_{P_Z}(\ell(w, Z) - L(w)) = 0$ for every $w$ so

$$\mathbb{E}_{P_Z Q_W}(\ell(W, Z) - L(W)) = \mathbb{E}_{P_Z}(\ell(W, Z) - L(W)) = 0.$$

We invoke Theorem 1 and substitute our expression for $\Phi_m$ to get

$$\text{gen} = \mathbb{E}_{P_{S,W}}\left(L(W) - \hat{L}(W, S)\right) \leq \frac{\sigma^2 \lambda}{2n} + \frac{1}{\lambda} \sum_{j=1}^{J} \mathbb{E}_{P_{S_j}} D\left(P_{W|S_j} \| Q_W\right).$$

The corollary then follows with the choice

$$\lambda := \sqrt{\frac{2n \sum_{j=1}^{J} \mathbb{E}_{P_{S_j}} D\left(P_{W|S_j} \| Q_W\right)}{\sigma^2}}.$$

## B DETAILED ARGUMENT FOR REMARK 4

In the $m = n$ version of the bound, the KL divergence $D\left(P_{W|S} \| Q_W\right)$ appears on the right hand side which is finite iff $P_{W|S} \ll Q_W$ (where $\ll$ denotes absolute continuity). Hence,

$$P_S\left(D\left(P_{W|S} \| Q_W\right) < \infty\right) = P_S(P_{W|S} \ll Q_W) = P_S\left(P_{W|S}(\{w : Q_W(w) > 0\}) = 1\right) \quad (20)$$

where the last equality holds since $P_{W|S}$ is a Dirac distribution and therefore $P_{W|S} \ll Q_W$ iff $Q_W(w) > 0$ for the single mass point $w$ of $P_{W|S}$. Clearly, in this example $P_W := \mathbb{E}_{P_S} P_{W|S}$ is a continuous distribution (i.e., it measures all singletons with probability 0) and $Q_W$ cannot have more than countably many points of positive mass, so by $\sigma$-additivity, we have

$$\mathbb{E}_{P_S}\left(P_{W|S}(\{w : Q_W(w) > 0\})\right) = P_W(\{w : Q_W(w) > 0\}) = 0 \quad (21)$$

Since the integrand on the left hand side of (21) is nonnegative, the integral being $0$ implies that the integrand has to be $0$ almost surely. Therefore, (20) and (21) together imply that $P_S(D(P_{W|S}\|Q_W) < \infty) = 0$, so in case $m = n$, the term $\mathbb{E}_{P_S} D(P_{W|S}\|Q_W)$ which appears on the right hand side is infinite regardless of the choice of $P_W$, making the bound vacuous.

Indeed, it can be seen from this argument that this property of the $m = n$ version of the bound is not specific to this example, but it applies whenever the algorithm $P_{W|S}$ is a Dirac distribution (i.e., the algorithm maps every training set $S$ deterministically to a hypothesis $W$) but the marginal distribution $P_W$ is continuous.

## C  CALCULATIONS FOR TABLE 1 IN SECTION 5

In this appendix, we provide the full derivations for the rates shown in Table 1.

We substitute (15) into (9) to obtain

$$\text{gen} \leq \sqrt{\mathcal{O}(n^{-2} J m^{\gamma})} = \mathcal{O}(n^{-\frac{1}{2}} m^{\frac{\gamma-1}{2}}).$$

The entry in the second column of the first row then follows by substituting $m = \Theta(n^{\alpha})$. For the second row, we need the following lemma.

**Lemma 2.** *If $r, s \in [0, 1]$, $\text{kl}(r\|s) \leq x$, and $r = \mathcal{O}(n^{-\varepsilon})$, then $s - r \leq 2x + \mathcal{O}(n^{-\varepsilon} \log n)$.*

*Proof.* We recall that due to the definition of binary KL divergence

$$x \geq \text{kl}(r\|s) = r \log \frac{r}{s} + (1 - r) \log \frac{1 - r}{1 - s}$$

or, equivalently,

$$\log(1 - s) \geq \frac{r}{1 - r} \log r + \log(1 - r) - \frac{x}{1 - r} - \frac{r}{1 - r} \log s.$$

Since $s \leq 1$ and assuming that $n$ is large enough so that $r \leq 1/2$, we can further bound the right hand side and get

$$\log(1 - s) \geq \frac{r}{1 - r} \log r + \log(1 - r) - 2x.$$

With equivalent term manipulations and using the facts that $1 - \exp(-t) \leq t$ for all $t \in \mathbb{R}$ and $\log(1 - r) \geq -r - r^2$ for all $r \in [0, 1/2]$, we obtain

$$s \leq 1 - \exp\left(\frac{r}{1 - r} \log r + \log(1 - r) - 2x\right)$$

$$\leq 2x - \frac{r}{1 - r} \log r - \log(1 - r)$$

$$\leq 2x - \frac{r}{1 - r} \log r + r + r^2$$

$$\leq 2x - \frac{r}{2} \log r + r + r^2.$$

The lemma then follows by subtracting $r$ on both sides and substituting the convergence behavior of $r$ on the right hand side. □

We substitute (15) into (8) and get

$$\text{kl}\left(\mathbb{E}_{P_{S,W}} \hat{L}(W, S) \| \mathbb{E}_{P_{S,W}} L(W)\right) \leq \mathcal{O}(n^{-2} J m^{\gamma}) = \mathcal{O}(n^{-1} m^{\gamma-1}).$$

The entry in the second column of the second row now follows by substituting $m = \Theta(n^{\alpha})$ and applying Lemma 2.

For the optimizing choice of $\alpha$ (which is the same in both rows), we note that the exponent in the second column is decreasing in $\alpha$ when $\gamma < 1$, so in this case the maximum choice $\alpha^* = 1$ is optimal. Similarly, when $\gamma \geq 1$, the exponent is nondecreasing, so the minimum choice $\alpha^* = 0$ is optimal. Finally, the entry in the last column follows by substituting the optimal $\alpha^*$ into the entry in the second column.

# D    PROOF OF THEOREM 2

We prove the theorem by first explicitly constructing $\ell$, $P_Z$, and $P_{W|S}$, and then proving that they satisfy the properties 1, 2, and 3 claimed in the theorem statement.

**Choice of $\ell$, $P_Z$, and $P_{W|S}$.**    We let $K \in \mathbb{N}$ be a parameter to be specified later, $\mathcal{Z} := [K]$, $\mathcal{W} := \{0,1\}^K$ (identified with the set of functions from $[K]$ to $\{0,1\}$), and
$$\ell(w,z) := w(z).$$
This means in particular that $\ell$ can only take the values 0 and 1 and hence is bounded as claimed in the theorem statement.

We define the sample distribution $P_Z$ as the uniform distribution over $\mathcal{Z}$. In preparation for specifying the learning algorithm $P_{W|S}$, we define the hypothesis $w_0$ via $w_0(k) = 0$ for all $k \in [K]$ and for any $z_1, \ldots, z_\ell \in \mathcal{Z}$, we define a hypothesis
$$\hat{w}_{z_1,\ldots,z_\ell}(k) := \begin{cases} 0, & \exists j \in [\ell] : k = z_j \\ 1, & \text{otherwise.} \end{cases}$$

Let $\phi \in [0,1]$ be another parameter to be specified later, and let $\Omega_\phi \subseteq \mathcal{Z}^m$ be such that $P_Z^m(\Omega_\phi) = \phi$. Choosing such a $\Omega_\phi$ is clearly possible as long as $\phi$ is an integer multiple of $K^{-m}$.

We can now define the randomized algorithm for our example as follows:
$$P_{W|S}(w|z_1,\ldots,z_n) := \begin{cases} \mathbb{1}_{w=w_0}, & (z_1,\ldots,z_m) \notin \Omega_\phi \\ \mathbb{1}_{w=\hat{w}_{z_{m+1},\ldots,z_n}}, & (z_1,\ldots,z_m) \in \Omega_\phi. \end{cases}$$

We conclude this part with a brief discussion of our choices that is intended to provide some intuitive understanding and a brief outline of the remainder of this proof. $w_0$ is a hypothesis of low loss, or more precisely, the empirical loss on any sample as well as the population loss will always have the lowest possible value 0. On the other hand, $\hat{w}_{z_1,\ldots,z_\ell}$ is a heavily overfitted hypothesis in the sense that it will have empirical loss 0 on any sample that is a subset of $\{z_1,\ldots,z_\ell\}$ but it has a population loss of at least $(K-\ell)/K$ which is close to 1 as long as $\ell \ll K$. The learning algorithm is chosen in such a way that most of the time it outputs $w_0$, and hence the generalization gap in this case will be 0. However, with a carefully calibrated probability $\phi$, it will output the hypothesis $\hat{w}_{z_{m+1},\ldots,z_n}$ which is overfitted to most of our training set, and consequently will exhibit a large generalization gap. The probability $\phi$ will be carefully chosen such that it is small enough to allow the divergence terms in (6) to vanish and at the same time is large enough so that (17) holds.

**Item 1: Assumptions of Theorem 1.**    What remains to be shown is that (5) is satisfied. This is straightforward as we have already argued the loss function $\ell$ is bounded in $[0,1]$. Hence, by Hoeffding's Lemma (see, e.g. (Boucheron et al., 2013, Lemma 2.2)), $\ell(w,Z)$ is $1/4$-subgaussian for all $w \in \mathcal{W}$ and therefore (5) is satisfied with
$$\Phi_m(\lambda) := \exp\left(\frac{\lambda^2}{8m}\right). \tag{22}$$

**Item 2: Choice of $Q_W$ such that the right hand side of (6) vanishes.**    We define
$$Q_W(w) := \alpha \mathbb{1}_{w=w_0} + (1-\alpha)K^{m-n} \sum_{z_{1:n-m} \in \mathcal{Z}^{n-m}} \mathbb{1}_{w=\hat{w}_{z_1,\ldots,z_{n-m}}}, \tag{23}$$
where $\alpha \in [0,1]$ is another parameter to be specified later.

Next, we show that the right rand side in (6) vanishes with appropriate choices for the parameters of our construction. Recalling the definition $S_1 := Z_{1:m}, \ldots, S_J := Z_{n-m+1:n}$, this means that we have
$$P_{W|S_1}(w|z_{1:m}) = \begin{cases} K^{m-n} \sum_{z_{m+1:n} \in \mathcal{Z}^{n-m}} \mathbb{1}_{w=\hat{w}_{z_{m+1:n}}}, & z_{1:m} \in \Omega_\phi \\ \mathbb{1}_{w=w_0}, & \text{otherwise,} \end{cases} \tag{24}$$
$$P_{W|S_j}(w|z_{1:m}) = (1-\phi)\mathbb{1}_{w=w_0} + \phi K^{2m-n} \sum_{z_{m+1:n-m} \in \mathcal{Z}^{n-2m}} \mathbb{1}_{w=\hat{w}_{z_{1:n-m}}} \tag{25}$$

for $j \in \{2, \ldots, J\}$.

With this we are ready to calculate bounds for the divergence terms that appear on the right hand side of (6). Conditioned on the event $S = z_{1:n}$ for some $z_{1:n} \in \mathcal{Z}^n$ with $z_{1:m} \in \Omega_\phi$ and denoting $\hat{\mathcal{W}} := \{w \in \mathcal{W} : P_{W|S_1}(w|z_{1:m}) > 0\}$, we have

$$
\begin{aligned}
D\left(P_{W|S_1} \| Q_W\right) &= \sum_{w \in \hat{\mathcal{W}}} P_{W|S_1}(w|z_{1:m}) \log \frac{P_{W|S_1}(w|z_{1:m})}{Q_W(w)} \\
&\overset{(a)}{=} \sum_{w \in \hat{\mathcal{W}}} P_{W|S_1}(w|z_{1:m}) \log \frac{K^{m-n} \sum_{z_{m+1:n} \in \mathcal{Z}^{n-m}} \mathbb{1}_{w = \hat{w}_{z_{m+1:n}}}}{(1-\alpha) K^{m-n} \sum_{z_{1:n-m} \in \mathcal{Z}^{n-m}} \mathbb{1}_{w = \hat{w}_{z_{1:n-m}}}} \\
&\overset{(b)}{=} \log \frac{1}{1-\alpha}
\end{aligned}
\tag{26}
$$

where step (a) follows by observing $w_0 \notin \hat{\mathcal{W}}$ and substituting (24) and (23) and step (b) by observing that the sums in the numerator and denominator are identical since they differ only in the names of the summation variables.

Next, we condition on the event $S = z_{1:n}$ for some $z_{1:n} \in \mathcal{Z}^n$ with $z_{1:m} \notin \Omega_\phi$. In this case, we obtain

$$
D\left(P_{W|S_1} \| Q_W\right) = \sum_{w \in \hat{\mathcal{W}}} P_{W|S_1}(w|z_{1:m}) \log \frac{P_{W|S_1}(w|z_{1:m})}{Q_W(w)} \overset{(a)}{=} \log \frac{1}{Q_W(w_0)} \overset{(b)}{=} \log \frac{1}{\alpha}
\tag{27}
$$

where step (a) follows because $W = w_0$ almost surely under $P_{W|S_1}(\cdot|z_{1:m})$ according to (24) and step (b) follows by substituting (23).

For $j \in \{2, \ldots, J\}$, we condition on the event $S = z_{1:n}$ for an arbitrary $z_{1:n} \in \mathcal{Z}^n$ and write $\hat{\mathcal{W}} := \{w \in \mathcal{W} : P_{W|S_j}(w|z_{(j-1)m+1:jm}) > 0\}$. Thus, we get

$$
\begin{aligned}
& D\left(P_{W|S_j} \| Q_W\right) \\
&= \sum_{w \in \hat{\mathcal{W}}} P_{W|S_j}(w|z_{(j-1)m+1:jm}) \log \frac{P_{W|S_j}(w|z_{(j-1)m+1:jm})}{Q_W(w)} \\
&= P_{W|S_j}(w_0|z_{(j-1)m+1:jm}) \log \frac{P_{W|S_j}(w_0|z_{(j-1)m+1:jm})}{Q_W(w_0)} \\
&\quad + \sum_{w \in \hat{\mathcal{W}} \setminus \{w_0\}} P_{W|S_j}(w|z_{(j-1)m+1:jm}) \log \frac{P_{W|S_j}(w|z_{(j-1)m+1:jm})}{Q_W(w)} \\
&\overset{(a)}{=} (1-\phi) \log \frac{1-\phi}{\alpha} + \sum_{w \in \hat{\mathcal{W}} \setminus \{w_0\}} P_{W|S_j}(w|z_{(j-1)m+1:jm}) \\
&\quad \cdot \log \frac{\phi K^{2m-n} \sum_{\hat{z}_{m+1:(j-1)m} \in \mathcal{Z}^{(j-2)m}, \hat{z}_{jm+1:n} \in \mathcal{Z}^{n-jm}} \mathbb{1}_{w = \hat{w}_{\hat{z}_{m+1}, \ldots, \hat{z}_{(j-1)m}, z_{(j-1)m+1}, \ldots, z_{jm}, \hat{z}_{jm+1}, \ldots, \hat{z}_n}}}{(1-\alpha) K^{m-n} \sum_{\hat{z}_{1:n-m} \in \mathcal{Z}^{n-m}} \mathbb{1}_{w = \hat{w}_{\hat{z}_1, \ldots, \hat{z}_{n-m}}}} \\
&\overset{(b)}{\leq} (1-\phi) \log \frac{1-\phi}{\alpha} + \sum_{w \in \hat{\mathcal{W}} \setminus \{w_0\}} P_{W|S_j}(w|z_{(j-1)m+1:jm}) \log \frac{\phi K^{2m-n}}{(1-\alpha) K^{m-n}} \\
&\overset{(c)}{=} (1-\phi) \log(1-\phi) + (1-\phi) \log \frac{1}{\alpha} + \phi \log \phi + \phi m \log K + \phi \log \frac{1}{1-\alpha} \\
&\leq \log \frac{1}{\alpha} + \phi m \log K + \phi \log \frac{1}{1-\alpha},
\end{aligned}
\tag{28}
$$

where step (a) follows by substituting (25) and (23), step (b) follows because the sum in the denominator contains all the summands that the sum in the numerator contains (hence it cannot be smaller), and in step (c) we use the fact that

$$
P_{W|S_j}(\hat{\mathcal{W}} \setminus \{w_0\}|z_{(j-1)m+1:jm}) = 1 - P_{W|S_j}(w_0|z_{(j-1)m+1:jm}) = \phi.
$$

We can now calculate a bound for the divergence terms in (6). Namely,

$$
\sum_{j=1}^{J} \mathbb{E}_{P_{S_j}} D\left(P_{W|S_j}||Q_W\right)
$$

$$
\stackrel{(a)}{=} P_{S_1}(\Omega_\phi)\mathbb{E}_{P_{S_1}}\left(D\left(P_{W|S_1}||Q_W\right)|S_1 \in \Omega_\phi\right) + (1 - P_{S_1}(\Omega_\phi))
$$

$$
\cdot \mathbb{E}_{P_{S_1}}\left(D\left(P_{W|S_1}||Q_W\right)|S_1 \notin \Omega_\phi\right) + \sum_{j=2}^{J} \mathbb{E}_{P_{S_j}} D\left(P_{W|S_j}||Q_W\right)
$$

$$
\stackrel{(b)}{\leq} \phi \log \frac{1}{1-\alpha} + (1 - \phi)\log\frac{1}{\alpha} + \left(\frac{n}{m} - 1\right)\left(\log\frac{1}{\alpha} + \phi m \log K + \phi \log\frac{1}{1-\alpha}\right)
$$

$$
\leq \phi \log \frac{1}{1-\alpha} + \frac{n}{m}\log\frac{1}{\alpha} + n\phi \log K + \frac{n}{m}\phi \log\frac{1}{1-\alpha}, \tag{29}
$$

where (a) is due to the law of total expectation and (b) follows by substituting (26), (27), and (28).

This allows us to bound the right hand side of (6) (from now on denoted as $\mathcal{R}$). Substituting (22) and (29), we obtain

$$
\mathcal{R} \leq \frac{\lambda}{8n} + \frac{\phi \log \frac{1}{1-\alpha} + \frac{n}{m}\log\frac{1}{\alpha} + n\phi \log K + \frac{n}{m}\phi \log\frac{1}{1-\alpha}}{\lambda}. \tag{30}
$$

With the choices

$$
K := 3n \log n
$$

$$
\alpha := \exp\left(-\frac{m}{n}\right) \tag{31}
$$

$$
\phi := \frac{3}{K} = \frac{1}{n \log n}
$$

$$
\lambda := \sqrt{n}
$$

and the observation

$$
1 - \alpha = 1 - \frac{1}{\exp\left(\frac{m}{n}\right)} > 1 - \frac{1}{1 + \frac{m}{n}} = \frac{m}{n+m}, \tag{32}
$$

we can further bound (30) and get

$$
\mathcal{R} \leq \frac{\sqrt{n}}{8n} + \frac{\frac{\log \frac{n+m}{m}}{n \log n} + 1 + \frac{\log(3n \log n)}{\log n} + \frac{\log \frac{n+m}{m}}{m \log n}}{\sqrt{n}}
$$

$$
\stackrel{(a)}{\leq} \frac{1}{\sqrt{n}}\left(\frac{1}{8} + \frac{\log(n+1)}{n \log n} + 1 + \frac{\log(3n \log n)}{\log n} + \frac{\log(n+1)}{\log n}\right)
$$

$$
= \frac{1}{\sqrt{n}}\left(\frac{17}{8} + \frac{\log(n+1)}{n \log n} + \frac{\log(n+1) + \log 3}{\log n} + \frac{\log \log n)}{\log n}\right),
$$

where in (a) we use the fact that since $m$ takes only values in $[n]$, we have $(n+m)/m \leq n+1$ as well as $m \geq 1$ for all valid choices of $m$. Since all terms in the bracket are nonincreasing, the bound (6) vanishes with rate $\mathcal{O}(1/\sqrt{n})$.

**Item 3: The generalization error does not vanish reasonably fast with small error probability.**
With probability $\phi = 1/n \log n$ over $P_S$, we have $W = \hat{w}_{Z_{m+1:n}}$ almost surely over $P_{W|S}$ and therefore

$$
\mathbb{E}_{P_{W|S}}L(W) - \mathbb{E}_{P_{W|S}}\hat{L}(W,S) \geq \frac{K - n + m}{K} - \frac{m}{n}
$$

$$
= \frac{3n \log n - n + m}{3n \log n} - \frac{m}{n}
$$

$$
= \left(1 - \frac{m}{n}\right)\left(1 - \frac{1}{3 \log n}\right).
$$

Towards a contradiction, we assume that the logical opposite of (17) holds, i.e., that we have $A_n$ which vanishes as $n \to \infty$ and $B_n$ which vanishes with order $o(1/f(n \log n))$ as $n \to \infty$, and

$$P_S \left( \mathbb{E}_{P_{W|S}} L(W) - \mathbb{E}_{P_{W|S}} \hat{L}(W, S) < A_n + B_n f \left( \frac{1}{\delta} \right) \right) > \delta$$

for a sufficiently large $n$ and all $\delta \in [0, 1]$.

Setting $\delta := \phi = 1/n \log n$, this implies that with probability at least $\delta$

$$
\begin{aligned}
A_n + B_n f(n \log n) &= A_n + B_n f \left( \frac{1}{\delta} \right) \\
&> \mathbb{E}_{P_{W|S}} L(W) - \mathbb{E}_{P_{W|S}} \hat{L}(W, S) \\
&\geq \left( 1 - \frac{m}{n} \right) \left( 1 - \frac{1}{3 \log n} \right).
\end{aligned}
$$

We can observe that since the left and right hand side of this inequality chain are not random and the entire chain holds with positive probability, the left hand side is a valid (deterministic) upper bound for the right hand side. Since $m/n$ vanishes, the right hand side converges to 1, so in particular it is clearly bounded away from 0. This contradicts the assertion that $A_n$ converges to 0 and $B_n$ converges to 0 with rate $o(1/f(n \log n))$.

## E    PROOF OF (19)

To see this, we can sum the instantaneous divergences conditioned on the overfitting event calculated in (26) and (28):

$$
\begin{aligned}
\Delta_{\text{inst}} &\leq \log \frac{1}{1 - \alpha} + \frac{n}{m} \left( \log \frac{1}{\alpha} + \phi m \log K + \phi \log \frac{1}{1 - \alpha} \right) \\
&\overset{(a)}{<} \log (n + m) - \log m + 1 + \frac{\log 3}{\log n} + 1 + \frac{\log \log n}{\log n} + \frac{\log \frac{n+m}{m}}{n \log n},
\end{aligned}
$$

where in step (a) we have substituted (31) and (32). Clearly the dominating term is $\log(n + m)$ which can be bounded as

$$\log(n + m) \leq \log(2n) = \log 2 + \log n.$$

