# OpenReview forum: "Block-Sample MAC-Bayes Generalization Bounds"
_ICLR.cc/2026/Conference — ICLR 2026 Poster_

### Official Review · Reviewer_vrZ6 · 2025-10-16

**Soundness:** 2
**Presentation:** 2
**Contribution:** 2
**Rating:** 4
**Confidence:** 2

**Summary:**

MAC-Bayes bounds (mean approximately correct)  are a  variation of PAC-Bayes bounds that are not necessarily valid with high probability but bound the expected generalization error.
The study considers „block-sampled“ MAC-Bayes bounds, in which the data is split into block/batches and the divergence term between prior and posterior is defined on block level.
The submitted paper studies the optimization of these „block-sampled“ MAC-Bayes bounds and whether they can be turned into
PAC-Bayes versions which improve on current PAC-Bayes results, which the authors show to be not possible in general.

**Strengths:**

I found the general topic of the study interesting.
I would like to stress that also the negative result is interesting.

**Weaknesses:**

* The „impossibility result“ in Theorem 2 can be viewed as a general form of the properly cited results by

Hrayr Harutyunyan, Greg Ver Steeg, and Aram Galstyan. Formal limitations of sample-wise
information-theoretic generalization bounds. In 2022 IEEE Information Theory Workshop (ITW),
pp. 440–445. IEEE, 2022.

* I was wondering: how are the results related to the results by

Recursive PAC-Bayes: A frequentist approach to sequential prior updates with no information loss
YS Wu, Y Zhang, BE Chérief-Abdellatif, Y Seldin
Advances in Neural Information Processing Systems 37, 17947-17971

which - in a different setting - consider a split of the training data and incrementally update the PAC-Bayes bound.

* Section 4: I was somehow not convinced by this example that should show the usefulness of the new bounds. The prior leading to (11) depends on the „batch size“ m. I accept that in PAC-Baysian analysis the prior is not necessary some form of „prior belief“ as in stared Bayesian analysis, but rather a tool to get tight performance guarantees. Still, that the prior depends on m does not feel right to me - where should such a prior come from? Could the authors discuss this in more detail?  The fact that the results are better than without blocks-sampling is then also heavily dependent on the prior (end of  Section 4). Why should this prior be used for the  m=n case? Is the prior real „prior belief“  here? If yes, where does it come from for the m-n case? If not - if it is a tool to get tight bounds- why this choice for n=m?


Minor comments:

* Authors forgot to introduce the prior Q_W when introducing PAC-Bayes in equation (1).
* Can the loss in the example 4 be viewed as some known robust loss function?

**Questions:**

Please see "Weaknesses" above.

---

> ### Author Response · Authors · 2025-11-19
>
> (In this comment, we only reply to the concerns/questions from Reviewer vrZ6. For an overview of changes made in the revised version of the paper, please see our top-level comment.)
>
> Thank you for your review and the detailed comments and questions which have helped us to strengthen our paper significantly. In the following, we give detailed responses to your questions and comments (Q1 = first bullet point, ..., Q5 = last bullet point):
>
> ### Q1/Q2.
> We are grateful for the additional literature reference of which we were not previously aware. The bound proposed by Wu et al is quite different from ours due to its recursive character, however, it definitely deserves a discussion in the literature section since this bound is also based on a separation of the training data into blocks. We have therefore added the reference towards the end of Section 2 (Related Works) -- you can find it by looking for the blue text color which we use to highlight revised parts of the paper.
>
> ### Q3.
> This is an excellent point which we believe has helped us strengthen the discussion of the example significantly. Indeed, the prior does not need to represent "prior belief" and can be chosen freely for the analysis (we have made sure to explain this point well in our revised version of the introduction). Therefore, we completely understand the concern that the m=n case being vacuous could potentially be the result of a bad choice of prior. However, we are able to argue that the m=n bound will be vacuous for *every possible* choice of prior. We have added Remark 4 at the end of Section 4 to make this point explicit in the paper (technical details of the mathematical argument are relegated to Appendix B due to the constraint on number of pages).
>
> ### Q4.
> Following this concern and also similar concerns raised by the other reviewers, we have almost completely re-written the introduction, taking care to explain in detail what the role of $Q_W$ is.
>
> ### Q5.
> It can certainly be seen as an example of a robust loss function since the truncation means that the effect of outliers in the training data on the overall fit of the model is dampened significantly. While it does not appear to be a very common robust loss, we were indeed able to find an example of the truncated square loss being applied in a practical ML algorithm (Le and Zach 2021) and have added the reference to Section 4 (directly after we recall the definition of the loss function).

---

### Official Review · Reviewer_7GiM · 2025-10-30

**Soundness:** 4
**Presentation:** 2
**Contribution:** 4
**Rating:** 8
**Confidence:** 4

**Summary:**

This paper presents a mean posterior error bound that relates
$$\mathbb{E} \\{ d( \rho\\{ \hat L_n \\}, \rho\\{ \mathbb{E} L_n \\})  \\}$$
to
$$\sum_{j=1}^J \mathbb{E}\\{  D_{\text{KL}}(\rho_j\\|\pi) \\} + J \cdot \text{``per block CGF''}$$
where $\mathbb E$ is over the randomness of the observations, $\rho$ is the data-dependent posterior probability, and $\pi$ a prior probability; importantly, the novelty of the paper lies in $\rho_j$, the posterior probability *after observing the block #j*.

This implies novel generalization bounds that scale as
$$\sqrt{ \frac 1 n  \sum_{j=1}^J \mathbb{E}\\{  D_{\text{KL}}(\rho_j\\|\pi) \\}}$$
where the usual ``complexity term'' is broken down linearly into block components.

**Strengths:**

I found the block decomposition concept novel, and the authors did convince me that the technique drives tighter bounds.

Some might argue that this is an interpolation between the individual-sample ($m=1$) and the bulk-sample ($m=n$) regimes (borrowing the leave-one-out analysis technique, equality (b), from the former). However, the resulting bounds are tighter as a result of this interpolation, with the "optimal block rate" clearly explored in Section 5.

**Weaknesses:**

For those outside the PAC-Bayes community, Sections 1-2 do not provide a good introduction to the concepts involved. In particular, *what is $Q_W$*? PAC-Bayes people of course know this is the prior probability; however, even if explicitly mentioned, the role of this prior would usually still be quite confusing for generic readers. The authors did even less in this regard by expending 0 words on $Q_W$, and how the evolution to $P_{W|S_j}$ is to be construed.

Very minor notation issue: $\mathbb E_{P_S}$ vs. $\mathbb E_{S}$.

**Questions:**

In general, how does the quantity

$$K_{n} := m^{-1}\mathbb{E} \\{ D_{\text{KL}} (\rho_m \\|\pi) \\}$$

evolve as a sequence of $m$? Here $m$ agrees with the definition in your paper (i.e. block size), $\pi$ is a prior probability, and $\rho_m$ is the posterior probability evolved from $\pi$ after seeing $m$ i.i.d. observations. I believe this is important to better conceptualize the benefits of the block decomposition in bounds e.g. (10). Some graphical exploration would be helpful.

---

> ### Comment · Reviewer_7GiM · 2025-11-14
> **Write-up**
>
> Look, all three of us complained unanimously about the write-up quality of the paper, in particular, some underexplained notations.
>
> Although, as a matter of personal taste, I liked the results of the paper and gave it a high rating, I do believe this paper needs a major write-up overhaul for publication at any major venue.

---

> > ### Author Response · Authors · 2025-11-19
> >
> > (In this comment, we only reply to the concerns/questions from Reviewer 7GiM. For an overview of changes made in the revised version of the paper, please see our top-level comment.)
> >
> > Thank you for your encouraging review and especially for your detailed and constructive criticism regarding the write-up quality of the introductory part of our paper, which have helped us to make very substantial improvements to the paper. We have taken your concerns in your initial review and the additional emphasis expressed in the additional comment very seriously and re-written most of the introductory part. We believe that this has made the paper much more readable and addressed your concerns, along with the similar concerns also expressed by the other reviewers. In the following, we give a detailed point-by-point response to your comments:
> >
> > ### Weaknesses:
> > We have re-written substantial parts of the introduction, taking special care to explain in particular the roles of the prior $Q_W$ and of the marginal distributions $P_{W|S_j}$ well. We have also checked the entire paper for occurrences of the notational inconsistency that was pointed out and replaced $E_S$ with $E_{P_S}$ throughout the paper.
> >
> > ### Questions:
> > We agree that this is the important term that determines the scaling behavior of the bound. For the example in Section 4, if we understand correctly it is eq. (13) divided by $m$, which comes out to be $1/2(n-m)$. Up to the square root and the leading factor of $\frac{1}{2}$, this is exactly the upper bound given in (14). For other learning scenarios, this needs to be evaluated with the specific example at hand on a case by case basis.

---

> > > ### Comment · Reviewer_7GiM · 2025-11-23
> > >
> > > Thanks for the updates. I’ll keep my ratings unchanged.

---

### Official Review · Reviewer_HaRN · 2025-11-07

**Soundness:** 3
**Presentation:** 1
**Contribution:** 2
**Rating:** 4
**Confidence:** 3

**Summary:**

This paper studies the MAC-Bayes bounds. They proposed a family of novel block-sample MAC-Bayes bounds, which generalized the MAC-Bayes bounds in the sense that it aggregates the contributions of a collection of disjoint subsets of the sample (on the MAC-Bayes bound) instead of consider the sample a whole. They proved the convergence of the block-sample bound and showed its superior characteristic power compared with the original PAC-Bayes bounds in a simple application-Gaussian mean estimation. At last, the paper discussed how the block size could affect the convergence and the possibility of transforming the block-sample bound into a high probability form.

**Strengths:**

* Although I did not verify all the proofs in detail, they appear to be sound overall.
* While not significant, the block-sample bound seems to be a non-trivial generalization of the prior results.

**Weaknesses:**

I think the authors should put more effort in the presentations of this paper because of the following reasons:
* The authors throw out the definition of PAC-Bayes bounds at the very beginning, however, without giving enough explanation for the term in the expression. For instance, I do not see any description of $Q_W$ and have trouble in understanding what it means. Also, I advise the author to give a concrete example of $I(n,d)$ in addition to just saying it is proportional to $n$ and $d$. Similarly, instead of just saying what $P_{W|S}$ does in general, give a concrete example to demonstrate it.
* The authors introduced the "individual-sample bound" in describing their first contribution but did not even explain what it is.
* From my perspective, the authors did a poor job in motivating the audiences on their contributions. The author only briefly mentioned that their result is a generalization of the MAC-Bayes bound and individual-sample bound (which they did not define), they did not emphasize the significance of their generalization. In particular, the authors did not well explain what advantages this generalization can bring upon the original MAC-Bayes bound in the contribution part.
* While the author gave an example on mean estimation of Gaussian in Section 4 to demonstrate the usefulness of the block-sample bound compared to the PAC-Bayes bound, I am not convinced by this example due to its overly simplicity. In particular, the PAC-Bayes bound exhibits a poor generalization for this application not because this task is hard. In fact, a simple Hoeffding bound can give a tight error bound on the mean estimation. Therefore, this example is not enough to demonstrate the significance of the generalization. With that being said, I still suggest to move this example into the introduction to give a more fluent and motivative presentation.
* In Section 3, it would be more accessible for the audience if the author could briefly explain the assumption in the main theorem in plain language and sketch their proof idea of the theorem in advance.

I will consider raise my rating if the authors address my concerns.

**Questions:**

* What does the assumption (3) in the main theorem implies intuitively?

---

> ### Author Response · Authors · 2025-11-19
>
> (In this comment, we only reply to the concerns/questions from Reviewer HaRN. For an overview of changes made in the revised version of the paper, please see our top-level comment.)
>
> Thank you for your constructive review and in particular for the many detailed comments and questions that have helped us to enormously improve the readability of our paper, in particular in the introductory part. We understand that the introductory part of our submitted paper contained flaws that made the paper unnecessarily hard to understand. We have taken your feedback along with similar concerns raised by the other reviewers very seriously and re-written most of the introduction. With this, we believe that we were able to address the concerns raised in your review. In the following, we give a detailed point-by-point response (Q1 = first bullet point, ..., Q6 = last bullet point).
>
> ### Q1.
> In the revised version of the introduction, we have made sure to explain all three of the quantities you mention (along with all the other quantities relevant to their understanding) in detail. We have also followed your advice to give a concrete example (a neural network used for image classification) and connect the theoretical concepts to this example explicitly.
>
> ### Q2.
> The individual-sample bound is discussed in the 4th paragraph of Section 2 (Related Works). We agree that it was not a good idea to mention this bound in the introduction before it is actually explained what it is. We have addressed this concern by removing the mentioning of this bound from the abstract and introduction and we have added slightly more detail where it is discussed in Section 2 (look for the blue color text towards the beginning of the 4th paragraph).
>
> ### Q3.
> We have added some text immediately before the contributions list of the introduction to address this concern. In it, we do the following:
> * We say explicitly that the main motivation for the new bounds is their promise to make PAC-Bayes type bounds tighter for certain learning algorithms
> * We introduce the Section 4 example (as suggested in Q4) and say explicitly what the advantage of the block-sample bound is in the case of that example.
>
> ### Q4.
> As suggested, we are now introducing the example already in the introduction (which has also helped us address Q3). While we do understand the concern that the example is too simplistic to conclusively argue that the block-sample bound will be useful for practical learning algorithms, we believe that the example still demonstrates that the new bound holds significant promise. Due to the many additional complications that arise when applying PAC-Bayes bounds to practical ML algorithms, it is common practice in the field that there are theoretical papers that introduce new types of bounds which are evaluated only for simple examples (e.g., Asadi et al. (2018), Bu et al. (2020), Hellström and Durisi (2020), Steinke & Zakynthinou (2020), Esposito et al. (2021)) and there are other works dedicated to applying such bounds to more practical ML problems (e.g., Langford & Caruana (2001), Dziugaite & Roy (2017), Perez-Ortiz et al. (2021)). We felt it similarly necessary to limit the scope of the present paper and relegate the application to practical algorithms to future research. With that being said, we certainly understand the concern that even given this limitation in scope, we could have done a better job in motivating the example and putting it into proper context. We believe that in the new paragraph in the revised version that introduced the example already in the introduction, we have addressed this concern by improving on explaining the motivation of the example and putting it in better overall context.
>
> ### Q5.
> We have added a brief sketch of the proof of Theorem 1 at the beginning of the proof and added Remark 1 after the proof which briefly gives an intuition for the main assumption of the theorem.
>
> ### Q6.
> It can be understood as a assuming a bound for the moment-generating function of the loss under the prior. This assumption is, e.g., satisfied whenever the loss is bounded or subgaussian. We agree that the readers of the paper will also benefit from the answer to this question and have therefore put this explanation into the newly added Remark 1 after the proof of Theorem 1.

---

> > ### Comment · Reviewer_HaRN · 2025-11-23
> >
> > We thank the authors for seriously considering our advice and believe the most recent revision has addressed most of our concerns. Although we still have some doubt on the significance of the paper's contribution, we would like to raise our rating.

---

### Author Response · Authors · 2025-11-19
**Summary of changes in the revised version**

We are very grateful to the reviewers for the time and effort they have invested to provide comments on our paper that have helped us to make substantial improvements to the paper. We have taken the concerns raised by all three reviewers regarding the quality of the writing, especially of the introductory sections, particularly seriously. In this post, we briefly summarize the main changes we have made to the paper (for convenience, all text that is new or changed is printed in blue in the uploaded revised version). For detailed responses to individual comments made by the reviewers, please see our comments posted on each of the reviews.
* We have re-written most of Section 1 to give a far more comprehensive and detailed introduction to the general topic of the paper, as well as to introduce and explain every notation used in full detail.
* In the course of this, we are now illustrating the notions of statistical learning theory with an image classification example to make the paper more approachable and the overall motivation clearer.
* We now also introduce the simple example from Section 4 already in Section 1.
* Because this means that almost all of the notations that were previously introduced in Section 2 are now fully defined in Section 1, we have chosen to delete the former Section 2 and define the few remaining notations where they first appear.
* To make the paper structure a bit more straightforward, we have promoted the former Section 1.1 (Related Works) to a new top-level section, forming now Section 2.
* We have added many details and additional discussions into the paper which we are confident comprehensively address the other concerns raised by the reviewers. For details, see our individual responses to the reviews below.

---

### Author Response · Authors · 2025-12-03
**Summary of review/rebuttal process and final remarks**

The constructive criticisms of the reviewers have helped us to substantially improve our paper which was already quite highly rated during initial review.

* All three reviewers have strongly emphasized the weaknesses in presentation of the paper in the introductory sections. In response, we have fully re-written these parts in the revised version. Two out of the three reviewers have confirmed that their concerns have been addressed with this change, one of them raising their score and the other one maintaining their already very positive rating of our paper. The third reviewer unfortunately did not have a chance to respond before discussions were cut short, but we are certain that we were able to fully address all concerns.
* Reviewer vrZ6 has pointed out a technical weakness in the baseline comparison of pur example. In revision, we have added an additional mathematical argument  (Remark 4 and Appendix B in the revised version) which we are certain turns this weakness into a strength. Unfortunately, the reviewer did not have a chance to respond before discussions were cut short, but we think that we have addressed this concern in a particularly convincing manner.
* We have also carefully addressed every minor issue raised by the reviewers during our revisions, please see our detailed point-by-point responses below.

We would also like to emphasize that the overall view of our paper was quite positive even in the initial reviews, in particular:

* The topic of the paper and nature of the results are of interest to the field (specially emphasized by reviewers 7GiM and vrZ6)
* Our results are novel (specially emphasized by all three reviewers)
* The paper is technically sound (specially emphasized by reviewers 7GiM and HaRN)
* Reviewer vrZ6 has stressed that the negative result contained in the paper is also of substantial interest.

---

### Meta-Review · Area_Chair_e5Nc · 2026-01-06

**Summary:**

This paper studies MAC-Bayes (mean approximately correct) bounds and introduces a new family of block-sampled MAC-Bayes bounds, which generalize classical MAC-Bayes by aggregating contributions from disjoint data blocks (subsets) instead of treating the sample as a whole. The authors prove convergence of the block-sample bounds and demonstrate their superior characterization power compared to standard PAC-Bayes bounds on an example of Gaussian mean estimation. Finally, the authors  study optimization of block-sampled MAC-Bayes bounds and establish that,  generally speaking, they cannot be transformed into improved high-probability (PAC-Bayes) bounds.

**Reviewer Concerns:**

The reviewers had a generally positive assessment of the paper, noting that the paper addresses an important problem, and is technically sound. They noted that block-sample bound seems to be a non-trivial generalization of the prior results, which leads to tighter bounds. They also noted adequate exploration of optimal block rate. One of the reviewers also found the negative result (on the possibility of PAC-Bayes bounds of a similar form) interesting.

The reviewers strongly highlighted issues with the paper’s presentation, and specifically the need for better motivation of the contribution as well as proper notations. The authors addressed those concerns in the rebuttal by reorganization of the paper which included extensive revision of the Introduction and other sections, and comprehensive definition/explanation of the used notations. One of the reviewers expressed concerns whether the example provided in the paper demonstrated the usefulness of the new bounds, which the authors addressed by providing additional mathematical arguments in Sec. 4 and Appendix B. The authors also clarified the difference between the present contribution and prior work (Wu et al, 2024). Overall, the authors were able to adequately address reviewer concerns.

**Reviewer Scores:**

HaRN: 4->6;
7GiM:  8 unchanged ;
vrZ6: 4->6

---

### Decision · Program_Chairs · 2026-01-26

Accept (Poster)